# Comparing random walks in graph embedding and link prediction

**Adilson Vital Jr.[1], Filipi Nascimento Silva[2]\*, Diego Raphael Amancio[1]**

**1** Institute of Mathematics and Computer Science, USP, São Carlos, SP, Brazil, **2** The Observatory on Social Media (OSoMe), Indiana University, Bloomington, Indiana, United States of America

\* filsilva@iu.edu

**Data Availability Statement:** Data employed in this paper is available from the Netzschleuder network repository https://networks.skewed.de.

**Funding:** DRA acknowledges financial support from CNPq (grant no. 311074/2021-9) and São

## Abstract

Random walks find extensive applications across various complex network domains, including embedding generation and link prediction. Despite the widespread utilization of random walks, the precise impact of distinct biases on embedding generation from sequence data and their subsequent effects on link prediction remain elusive. We conduct a comparative analysis of several random walk strategies, including the true self-avoiding random walk and the traditional random walk. We also analyze walks biased towards node degree and those with inverse node degree bias. Diverse adaptations of the node2vec algorithm to induce distinct exploratory behaviors were also investigated. Our empirical findings demonstrate that despite the varied behaviors inherent in these embeddings, only slight performance differences manifest in the context of link prediction. This implies the resilient recovery of network structure, regardless of the specific walk heuristic employed to traverse the network. Consequently, the results suggest that data generated from sequences governed by unknown mechanisms can be successfully reconstructed.

## Introduction

Network science has emerged as an interdisciplinary field for modeling and understanding complex phenomena across diverse domains. Many intricate systems can be effectively represented and modeled through complex networks, this includes biological and physical systems [1], textual content [2–5], epidemics [6], human mobility [7], among others [8]. Characterizing the elements of such systems—nodes and edges—are a crucial task in the field, and many measures have been developed to account for that [8]. These measures are vital in executing various tasks such as classification and prediction, particularly when dealing with real data that may contain partial, missing, or noisy information in the network.

Prediction of network edges requires the adoption of metrics and models that capture not just local pairwise relationships, but also the local and global characteristics of the network. Nodes within the same community, for example, may exhibit similar patterns and are more likely to connect [9]. Furthermore, network models that encapsulate the dynamics of real-world systems, such as the BA, geographic, or gravity models, can be harnessed to predict connections [8]. The prediction task essentially involves determining the likelihood that two

Paulo Research Foundation (FAPESP grant no. 2020/06271-0). FNS thanks AFOSR \#FA9550-19-1-0391 for the financial support.

**Competing interests:** The authors have declared that no competing interests exist.

distinct nodes connect [10], which can be derived from a similarity or distance metric associated with nodes embedded in a particular space.

Many different techniques have been designed to embed nodes from a network to a space in which distances or similarities can be defined [11], including approaches such as LLE [12], Laplacian Eigenmaps [13], and graph factorization [14]. Nonetheless, these traditional methods often struggle with large graphs due to costly calculations and approximations that compromise their efficacy. Recently, neural network embeddings of graphs have emerged as an excellent trade-off between performance and computational speed for link prediction tasks [15, 16].

Neural network graph embeddings aim to represent the network in a lower-dimensional space while preserving key attributes such as proximity and community structure [17, 18]. A few successful examples of incorporating random walks and embeddings in downstream tasks include Deepwalk [19], LINE [20], Grarep [21], node2vec [22], and struc2vec [23].

Despite the successes, several open questions persist, particularly regarding the optimization of these models and the role of random walks in information extraction from networks. Previous work has compared different random walks for knowledge acquisition [24] and network reconstruction tasks [25], among others [26]. Our research primarily focuses on the information extraction process from node sequences generated by random walks. We primarily explore two questions: (1) Do different random walks exhibit distinct performances when used as input to an embedding model in a link prediction task? (2) Do different walks extract similar node similarity information from the network? We investigated these research questions by employing four established classical random walk techniques [25]. We used the traditional random walk (RW), the true self-avoiding walk (TSAW), degree-biased walk (DG), and inverse-degree-biased walk (ID). Furthermore, we explored five distinct node2vec parameter settings across a total of 37 networks.

Our findings indicate that different walks perform similarly, exhibiting only a slight 3% and 4% difference in terms of AUC and AUC PR, respectively, when assessing the median link prediction performance. The pattern persists even when different walks are compared within the same graph. Regarding node similarities, embeddings generated by different walks displayed a robust positive Pearson correlation. Our dataset revealed the lowest correlation (0.85) between the degree-based walk (DG) and inverse-degree-based walk (ID), two walks operating on opposing principles.

Our findings suggest that graph embedding techniques can reconstruct the network structure independently of the walk heuristic. This insight is valuable when reconstructing networks from datasets where only sequences (walks) are known. This is the case of textual or mobility data. In these examples, the underlying network structures can potentially be recovered from sequences of words or geographical changes, regardless of the specific walk method used.

## Related works

Several random walk techniques have been documented in the last years [8]. Among these, DeepWalk [19] is a model known for its ability to acquire latent representations of nodes that encode structural information within a network, enabling their utilization in statistical models. DeepWalk leverages the conventional random walk methodology, wherein the selection of the next node is based on an equal probability among the neighboring nodes of the current node.

Network traversal produces sequences of nodes analogous to textual sentences composed of word sequences. In the training process, the Skip Gram neural network architecture is employed, wherein a one-hot vector representing a word is given as input, and the network

aims to predict the surrounding words within a specified window. Computationally, evaluating the probability function for a single node against the others incurs significant expense. To address this, an optimization technique involving hierarchical softmax was proposed. This approach constructs a binary tree of probabilities, leading to a reduction in computational complexity to the logarithm of the network size ($O(\log(|V|))$) for calculating only the probabilities along the path from the root to the leaf.

Asynchronous stochastic gradient descent (ASGD) has been used as the optimization method to improve scalability for large networks. Some works demonstrated outstanding performance when applied to multi-label classification tasks, surpassing alternative approaches such as SpectralClustering [27], EdgeCluster [28] and Modularity [29]. DeepWalk proved to be a very efficient framework, being easy to scale for large networks and resulting in great prediction results. Particularly in this work, we adopted the same optimization strategy.

With the same goal as DeepWalk, LINE [20] is a network embedding model designed to transform extensive information networks into low-dimensional embedding vectors. Its approach focuses on preserving both local and global structural information by leveraging first and second-order proximities. In unweighted networks, the first-order proximity simply indicates whether nodes are connected or not, represented as binary values (0 or 1). In weighted networks, the first-order proximity encompasses the weight of the connection, reflecting the strength or intensity of the link between nodes ($w_{uv}$).

Consider a scenario where two nodes lack a connection, resulting in a first-order proximity value of zero. To address this situation, the second-order proximity is leveraged by taking into account the presence of common neighbors between the nodes. Let us denote the set of first-order similarities for node $u$ as $p_u = (w_{u1}, w_{u2}, \ldots, w_{u|V|})$, and likewise, $p_v$ represents the set of first-order similarities for node $v$. Second-order proximity, in this case, refers to the indirect similarity between nodes u and v, calculated by the dot product between $p_u$ and $p_v$, and its magnitude increases with a greater number of shared connections between the two nodes.

Similar to DeepWalk, LINE encounters computational complexity issues while calculating probabilities. As a solution, it utilizes the Negative Sample technique [30] which instead of calculating the probability for all $|V|$ nodes, each positive edge is compared against a negative sample produced by a noise distribution having a size of $K$ that is proximate to 10. Moreover, LINE proved to be very efficient in tasks such as word analogy, document classification, link prediction, and network visualization, being able to scale for large networks. Compared with DeepWalk, the probability approximation method is even faster since it does not calculate a path on a tree but just a few negative examples. Also, the walk is not just based on the node connections but on the first and second-order proximity, improving local and global structural awareness of the embeddings.

One of the most well-known walk-based graph embeddings is node2vec [22]. It combines the pros of the two previously presented models achieving state-of-the-art performances in many link prediction and node classification tasks. One of the main contributions of this model was the usage of two parameters that control the bias to walk through the networks and collect the node sequences being possible to seek node homophily and structural equivalence.

Node homophily, arising from social relationships, posits that individuals belonging to the same community exhibit a tendency to be correlated and associated, so nodes from the same community should be embedded closely. In terms of structural equivalence, nodes with identical roles within a network, such as hubs, bridges, and isolated nodes, are expected to have short distances between them. It is important to note that these nodes can be located in different parts of the network but still maintain the same functional role.

In order to incorporate this information into the embedded vectors, the node2vec algorithm combines two well-known search strategies: depth-first sampling (DFS) and breadth-

first sampling (BFS). The inclusion of DFS in node2vec enables it to exhibit exploratory behavior by traversing the network in depth and preventing redundant visits to the same node. This capability allows node2vec to effectively map communities while being cautious about homophily.

Conversely, BFS, due to its local behavior centered around the starting node, provides the ability to effectively map the structural role of nodes. Similar to DeepWalk, node2vec employs Skip Gram and, like LINE, uses Negative Sampling to minimize computational costs. Additionally, it uses asynchronous Stochastic Gradient Descent (ASGD) as the optimization algorithm. Ultimately, node2vec represents a scalable model that outperforms other walk-based and classic models across various tasks, offering an efficient architecture with enhancements on the utilized walk bias.

Regarding traditional random walk, similar studies compare random walks in terms of node coverage speed. In [24], the authors analyzed four random walks when applied to different artificial network models to evaluate the learning curve and knowledge acquisition tasks. The walks used were the traditional random walk (RW), the degree-biased walk (DG), the inverse-degree biased walk (ID), and the true self-avoiding walk (TSAW), and the four network models were Erdős–Rényi, Barabási-Albert, Waxman, and Modular Networks.

The comparison focused on the percentage of network coverage achieved at each walking step. In this task, the results indicated that the TSAW outperformed other walks by reaching unknown nodes faster, followed by the traditional random walk, while both DG and ID walks exhibited the poorest performances, irrespective of the network model. The key distinction was in the efficiency of acquiring knowledge through the exploration of new nodes.

Another experiment involved comparing the random walks in the context of network reconstruction [25]. Using the network models from [24, 31], the same set of random walks was employed to traverse the network and gather sequences of visited nodes. The adjacency information of these nodes in the sequences was utilized to reconstruct a new graph. As the walk progressed, various structural metrics were collected, including the number of nodes, edges, average degree, average shortest path length, average clustering coefficient, and degree assortativity.

As the length of the random walk increased, various metrics were collected and machine learning models were trained to predict both the specific walk and network model used. The results showed that with shorter sequences, it was possible to achieve good accuracy for predicting the walk, whereas predicting the network models required longer sequences. Using a random forest model, an accuracy of 78.1% was achieved for predicting the network model with a sequence length of 500 steps, while an accuracy of 98.6% was acquired for a sequence length of 5000 steps. In terms of predicting the random walk used, a 72.9% accuracy was achieved with a sequence length of 500 steps, increasing to 98.8% accuracy for a sequence length of 5000 steps. These findings demonstrate the potential to reconstruct both the network and the random walk that generated a given sequence of nodes.

In contrast to previous studies, this paper explores the characteristics of unbiased and biased random walks in generating embeddings. The sequence of visited nodes is used to train an embedding model resulting in the creation of vector representations. The resulting embeddings are then applied to the link prediction task and the performance of different random walks is compared.

## Methodology

We leveraged random walk algorithms to gather node sequences from graphs. For these graphs, after preprocessing we intentionally removed a fraction of the edges, and taking the same amount of non-edges we created a labeled dataset with positive and negative edges

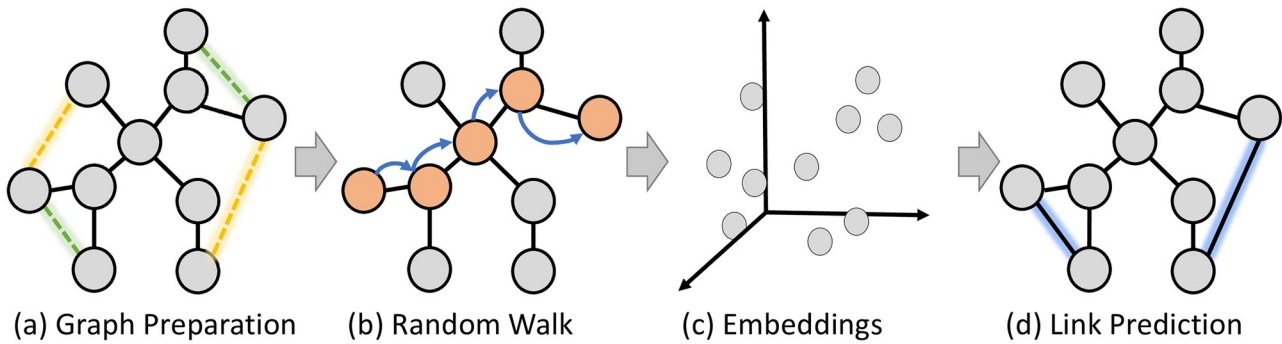

**Fig 1. Methodology used in the work.** The main steps applied are a) *graph preparation*, which includes graph selection and preprocessing; b) *random walk* application and *node sequences* collection; c) *node embeddings* process through the Skip-Gram model; and d) *link prediction* and similarity analysis.

samples respectively. With the gathered sequences, we utilized a skip-gram model to generate node embeddings, designed to preserve the inherent similarities among nodes. These embeddings were then employed to predict the missing edges in a link prediction task, thereby addressing our two principal research questions. Our primary objective was to probe the effects of varying the walk biases on link prediction performance and to compare the similarities across different walks. The methodology employed in this paper is illustrated in Fig 1. It comprises the following steps:

1. *Graph Preparation*: we divided the first step into two parts: graph selection and preprocessing. For the graph selection, we opted for diverse graphs that exhibit various structural properties and sizes. As for the preprocessing, we randomly removed a portion of the edges from each graph to form our positive sample, while an equal number of non-edges were selected as the negative sample. Finally, we designated the giant component of each graph as the final selected graph

2. *Random walk*: we implemented and applied nine distinct random walk strategies to generate node sequences for each graph: traditional Random Walk (RW), Degree-Based Random Walk (DG), Inverse Degree-Based Random Walk (ID), True Self-Avoiding (TSAW), and node2vec with five different parameter settings.

3. *Embeddings*: We utilized the node sequences as input for an embedding model to generate the node embeddings. For the four classic network walks (RW, DG, ID, TSAW), we employed the Word2Vec framework, while for Node2Vec, we utilized Skip-Gram. The hyperparameters of the embedding model were set according to the specifications in [19].

4. *Link Prediction*: The links between nodes are predicted by leveraging the similarities of their embeddings. To assess the performance of each random walk in the link prediction task, we evaluated ROC AUC (AUC) and Precision-Recall AUC (AUC PR). Additionally, we analyzed the distribution of embedding similarities produced by the different random walk strategies.

## Dataset

The graph selection process was carried out utilizing the Netzschleuder network repository [32]. Specifically, we constrained our selection to only unweighted, undirected, and unilateral networks within a node size range of 150 to 5000. We placed an emphasis on the

**Table 1. Graphs names and the number of nodes and edges for the utilized networks.**

| Graph Name | Nodes | Edges |
|---|---|---|
| Power | 3787 | 4423 |
| Interactome Vidal | 2460 | 4468 |
| Kegg Metabolic Hsa | 1784 | 4318 |
| Urban Streets Cairo | 1286 | 1530 |
| Collins Yeast | 925 | 6038 |
| Kegg Metabolic Lmo | 893 | 1984 |
| Copenhagen Fb Friends | 795 | 4814 |
| Dnc | 766 | 7787 |
| Copenhagen Bt | 689 | 59648 |
| Wiki Science | 663 | 4885 |
| Kegg Metabolic Buc | 576 | 1149 |
| Celegans Metabolic | 448 | 1519 |
| Sp Colocation Sfhh | 403 | 55168 |
| Eu Airlines | 401 | 2215 |
| Urban Streets London | 399 | 490 |
| Ugandan Village Friendship 16 | 369 | 1029 |
| Ugandan Village Friendship 8 | 368 | 1315 |
| Sp High School Proximity | 327 | 4364 |
| Ugandan Village Friendship 4 | 320 | 1557 |
| Facebook Friends | 316 | 1461 |
| Malaria Genes Hvr 1 | 305 | 2109 |
| Malaria Genes Hvr 5 | 296 | 2013 |
| Kegg Metabolic Uur | 284 | 533 |
| Urban Streets Paris | 271 | 333 |
| Malaria Genes Hvr 8 | 266 | 2949 |
| Contact | 248 | 1593 |
| Urban Streets New York | 239 | 314 |
| Ugandan Village Health Advice 11 | 220 | 395 |
| Sp Colocation Invs15 | 219 | 12544 |
| Ugandan Village Friendship 1 | 198 | 411 |
| Jazz Collab | 195 | 2057 |
| Sp High School New 2012 | 180 | 1665 |
| Sp High School Facebook | 156 | 1078 |
| Urban Streets San Francisco | 152 | 201 |
| Urban Streets Brasilia | 135 | 147 |
| Interactome Pdz | 126 | 145 |
| Student Cooperation | 124 | 183 |

connectedness of the nodes and the diversity of the network origins including social, transportation, biological, and other types. From an initial pool of 45 networks meeting the selection criteria, 8 were excluded due to having an insufficient number of edges to predict, resulting in a final set of 37 networks. The statistics of the used networks after preprocessing are provided in Table 1.

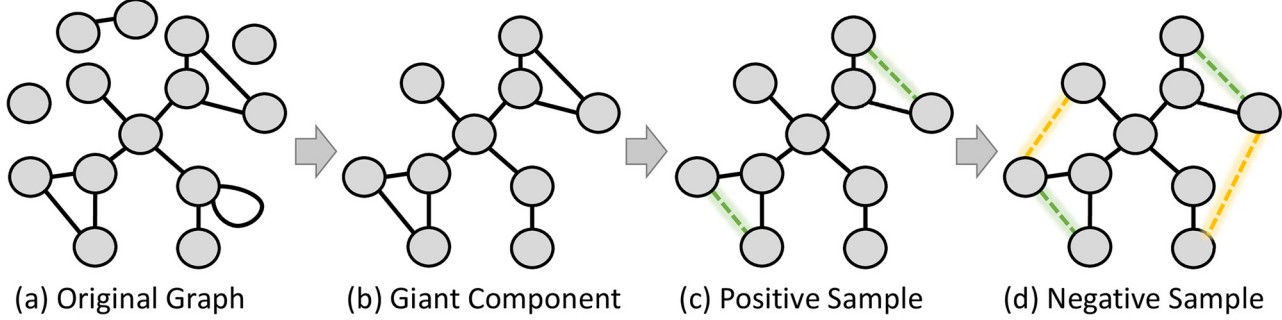

**Fig 2. Example of the preprocessing steps applied to the networks.** We began with the original graph (a) and used the largest connected component (b). Next, we removed a small portion of edges to create the labeled dataset (c). The removed edges, which represent the positive samples, are indicated by the green-dashed lines. To complete the final dataset, an equal number of non-edges (i.e., negative edges sample) are randomly selected and represented as dashed yellow lines in (d).

## Preprocessing

The networks underwent a series of preprocessing procedures, as illustrated in Fig 2. To ensure the connectedness of the graphs, we used only the largest connected component and removed self-edges, as demonstrated in Fig 2.

We generated a labeled dataset comprising an equal number of positive edges (existing edges that were removed) and negative edges (non-existing edges). This was achieved by randomly selecting and removing 25% of the edges from the network resulting in the formation of the first positive-edges sample (indicated by the green-dashed edges Fig 2(c). The same amount of negative edges was then chosen to compose the negative edges dataset. Finally, we used the same amount of positive edges to randomly select non-edges and label them as our negative samples, ending with the same number of samples for each class. The complete dataset containing positive and negative edges is depicted in Fig 2(d). Positive edges are represented by green dashed lines, while negative edges are indicated by yellow dashed lines.

## Random walk

A random walk is a sequence of nodes visited during a random traversal of a network. While this paper focuses on random walks with a finite number of steps, in general, the sequence of nodes can be infinite. Therefore, in our context, a random walk involves a starting node, a probability function, and a maximum walk length. The probability function employed to calculate the probability of transitioning from node $u$ to $v$ can incorporate various types of bias. The main features of random walks used in this paper are described below:

- *Starting node*: Two main techniques can be employed to determine the starting nodes for generating a set of random walks. The first technique involves predefining a maximum number of sequences and randomly selecting a starting node each time a new walk is initiated. Probabilities can be uniformly assigned or based on node attributes, such as node degree [33]. The second technique, employed in this paper, involves fixing a number of walks per node, denoted by $\beta$, where we used 40 walks per node as the starting node.

- *Probability function*: Different probability functions can be employed to establish the transition probability between node pairs in a random walk. The transition probability $P(v|u)$ of a node $u$ at iteration $i_t$ to transition to a specific neighbor $v$ in the subsequent iteration $i_{t+1}$ is

determined proportionally by the transition weight $\tau_{uv}$:

$$P(i_{t+1} = v \mid i_t = u) = \begin{cases} \tau_{uv}/Z_u, & \text{if } v \in \Gamma(u) \text{ and } Z_u = \sum_{z \in \Gamma(u)} \tau_{uz} \\ 0, & \text{otherwise} \end{cases} \tag{1}$$

To determine the probability, we normalize the transition weight $\tau_{uv}$ using the normalization factor $Z_u$, which is the sum of all transition weights from node $u$ to its neighbors in $\Gamma(u)$. In this paper, nodes that are not connected to node $u$ are always assigned a probability of zero. However, other techniques, such as a random walk with restart [34, 35] may assign a probability of returning to the initial node. Other techniques such as Lévy flight [36] can jump into a random node.

- *Maximum walk length*: For the maximum length of the walk, denoted by $\alpha$, some studies have utilized a variable number proportional to the node degree in order to limit the walk length and facilitate the mapping of hubs and well-connected nodes in large-scale networks [37]. However, other works have employed a fixed number, demonstrating that a lower number may not be sufficient to capture all the information necessary for optimal performance [22]. Additionally, a higher number of walks can lead to increased computational costs and memory usage. Studies suggest that performance is limited for walk lengths between 10 and 100, with performance improving as the length increases up to around 150 [22]. To obtain a stable value, we chose $\alpha$ to be 200. The dimension of the set of walks is the product of the total number of walks and the walk length, denoted as $(|V| \cdot \beta) \times \alpha = (|V| \cdot 40) \times 200$, and is dependent on the number of nodes per graph.

We employed four types of random walks. The first type utilizes only the number of neighbors as bias. The next two types use the network structure as bias, with one being the inverse of the other. Finally, the last type incorporates the memory of visited nodes as an attribute parameter for probability calculation. Besides the four classical walks, we also utilized five variations of the node2Vec framework. The classical random walks used are:

1. *Traditional random walk (RW)*: This is the simplest (unbiased) and most well-known random walk [38–40]. In this approach, the probabilities of transitioning to neighboring nodes are distributed uniformly, and the probability of transitioning from a node $u$ to a node $v$ is simply the inverse of the degree of node $u$, in other words, each neighbor will have a transition weight of 1:

$$\tau_{uv} = 1. \tag{2}$$

2. *Degree biased random walk (DG)*: The degree has been used as a walk bias to discern the nodes that possess greater importance in terms of their connectivity [37, 41]. In this work, we have adopted a strategy where the transition weight is directly proportional to the degree of neighboring nodes:

$$\tau_{uv} = |\Gamma(v)|. \tag{3}$$

By following this approach, nodes with higher degrees have a higher probability of being selected as the next node to traverse, thereby increasing the likelihood of identifying hubs as opposed to less connected nodes.

3. *Inverse degree-biased random walk (ID)*:
   An alternative approach to the degree-biased random walk can be achieved by utilizing the

degrees of neighboring nodes to limit the exploration of hubs and prioritize nodes with lower degrees. The transition weights in this strategy are computed as:

$$\tau_{uv} = |\Gamma(v)|^{-1}. \tag{4}$$

Previous studies have shown analytically that the inverse degree-biased walk outperforms other degree-biased walks in terms of time efficiency when exploring nodes in a network [41–43].

4. *True self-avoiding random walk (TSAW)*: In contrast to other random walks, the True Self-Avoiding Random Walk (TSAW) exhibits a nonstationarity probability function. The probability of transitioning from node $u$ to $v$ may vary through the course of the walk, requiring recalculation of the probabilities at each step. This phenomenon occurs because the transition probability in TSAW depends on the frequency with which a node is visited, which changes at each new walk step. As a result, TSAW is the most computationally expensive in terms of both memory and time. Two parameters are used for its calculations, the decay factor $\lambda$ and the frequency $f_v$ that node $v$ was visited [44, 45]. Because of the decay factor, the probability of transitioning to $v$ decreases while increasing the frequency at which $v$ is visited. In this study, we use a decay factor $\lambda = \ln(2)$ [24]. The transition weight is computed as

$$\tau_{uv} = e^{-\lambda f_v}. \tag{5}$$

where $f_x$ refers to the absolute number of times node $x$ has been visited.

For unvisited nodes, the numerator in Eq 5, $e^{-\lambda f_v} = 1$. This value subsequently decreases by half with each new visit to that node. i.e. it becomes 0.5, 0.25, and so on for 1 and 2 visits, respectively. This exploratory behavior results in TSAW achieving the best learning curve, outperforming other walk biases in terms of new nodes visited per iteration [24].

5. *Node2Vec*: Node2Vec is an algorithmic framework inspired by a combination of breadth-first search (BFS) and depth-first search (DFS) [22]. In BFS search, the walk starts with the root tree and aims to explore all nodes at the current depth before moving to the next depth level until reaching the leaves. On the other hand, DFS seeks to delve as deep as possible until it needs to backtrack and explore other unvisited paths.

When comparing BFS and DFS, the former exhibits a local and cautious behavior, whereas the latter acts as an explorer. Drawing inspiration from these characteristics, Node2Vec aims to incorporate both behaviors into the transition probabilities. The probability of visiting the last visited node will be determined by a return parameter $p$. A lower value of $p$ increases the likelihood of a local walk, which tends to stay close to the initial node. Nodes that are not the last visited node or not directly connected to it are considered outward nodes, with probabilities determined by an in-out parameter $q$. A lower value of $q$ corresponds to a shallower walk, increasing the chances of outward exploration. The neighbors of the latest visited node will have a fixed transition probability equal to 1, independent of the values of $p$ and $q$. In Fig 3, we can observe that the current node $i_t$ can transition back to $i_{t-1}$ with a transition weight of $1/p$, move to a neighbor of the latest visited node with a weight proportional to 1, or explore outward with a weight of $1/q$.

In order to compare different scenarios, we employed various combinations of $p$ and $q$. As a nomenclature convention, we will refer to these combinations as N2V(p, q). The first combination is N2V(1, 1), which serves as a baseline for comparison with the traditional random walk (RW). We also utilized N2V(1.5, 0.5), representing an explorer behavior that

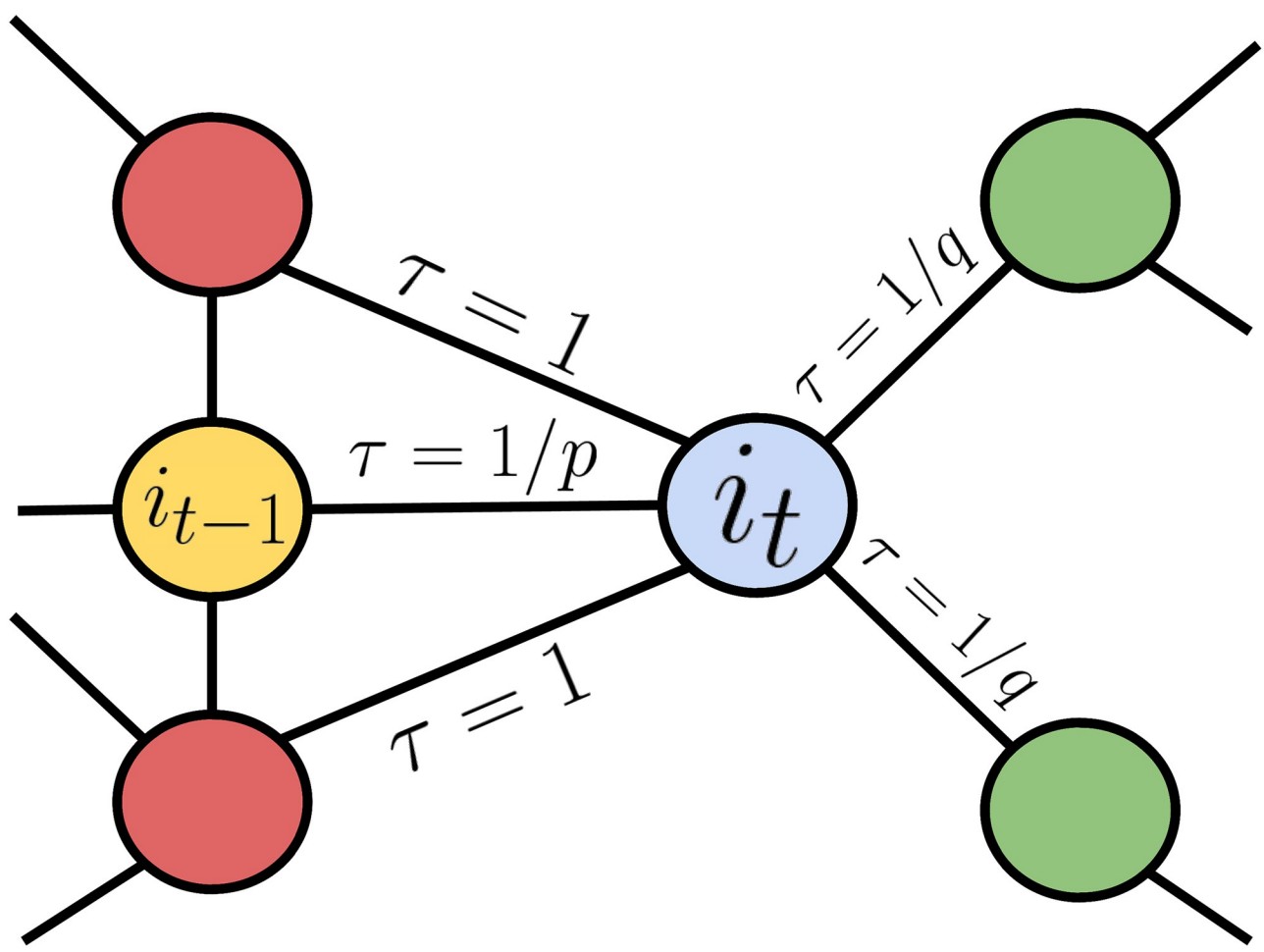

**Fig 3. Example of transition weights for the Node2Vec over a simple subgraph.** We can see the current blue node in the iteration $i_t$, being possible to return to the previous visited yellow node $i_{t-1}$, or continue an outward walk through the green ones.

avoids visited nodes, akin to TSAW. Furthermore, we considered a local walk, the inverse of the previous one, referred to as N2V(0.5, 1.5). Finally, by combining the two behaviors, we obtained N2V(2.0, 1.5) and N2V(0.25, 0.5).

## Node embedding

To analyze the four classical walks, we first gathered the node sequences. Next, we applied the Skip-Gram model within the Word2Vec framework, treating the sequence of visited nodes as a sequence of words, and generating their vector embeddings. After generating the node sequences using a predefined walking strategy, we again employed the Skip-Gram model to create node embeddings.

The primary objective of this node embedding phase is to transform the graph's nodes into a data structure capable of embedding information from the surrounding nodes and the graph structure. This transformation aims to represent each node $v \in V$ in a dense vector space of dimension $d$, resulting in a mapping $G = (V, E) \to \mathbb{R}^{|V| \times d}$, where $|V|$ is the number of nodes in the graph and $d$ is the dimensionality of the embedding vectors.

The Word2Vec [30, 46] is a deep learning model imported from natural language processing. This method is able to transform words into vectors, embedding their semantic and syntactic information, and allowing algebraic operations like addition and subtraction. Each word is initially embedded as a one-hot encoding, used as input and output in the training phase. One-hot vectors have the length of the vocabulary size, with all values being zero except the column of the corresponding word, which will be one.

The Word2Vec model can employ two network architectures to learn the final embedding: the continuous bag-of-words (CBOW) and the Skip-Gram architecture. In CBOW, the model is trained using the surrounding words as inputs and the central word as the expected output. The name "bag-of-words" refers to the fact that similarly to the traditional bag-of-words approach [47], the order and distance of the surrounding words from the central word do not have any impact on the final embedding.

In the Skip-Gram architecture, used in this paper for all walk types, the input is the central word, and the expected output is the surrounding words. Increasing the context window can improve the performance of the model, but it also requires more computational power. To account for the decreased relevance of more distant words, Skip-Gram assigns less weight to those words by sampling them less frequently during the training phase.

During the skip-gram model, the goal is to maximize the average logarithmic probability for every sequence of words $w_1, w_2, \ldots, w_T$:

$$\frac{1}{T}\sum_{t=1}^{T}\sum_{-c\leq j\leq c, j\neq 0}\log p(w_{t+j}|w_t). \tag{6}$$

Here $p$ represents the conditional probability function, $w_t$ the context, $c$ is the context size and $T$ denotes the length of the sentence.

The probability distribution $p(w_o|w_i)$ for word embeddings is initially calculated by applying the softmax function to the vector representations ($w$) of the input $i$ and output $o$ words. These vector representations are obtained from the one-hot vectors used to train the neural network and learn the embedding representation. $p(w_o|w_i)$ is computed as

$$p(w_o|w_i) = \frac{\exp(v'^T_{w_o}v_{w_i})}{\sum_{w=1}^{W}\exp(v^T_w v_{w_i})}, \tag{7}$$

where $W$ is the vocabulary size, $v_w$ is the vector representation of word $w$, $v_{w_i}$ is the vector of the input word $w_i$, and $v'_{w_o}$ is the produced output vector. Computing the denominator of this function, which involves all words in the vocabulary $W$, is computationally infeasible. To address this challenge, negative sampling has been proposed as a solution [30]. Negative sampling aims to distinguish the target word $w_o$ from noise distribution words by replacing the softmax function with a logistic regression calculation that involves only $k$ negative samples for each word. This significantly reduces the computational requirements during training, as $k$ can range from 2-5 for large datasets and 5 to 20 for small ones, which is substantially lower than the size of the vocabulary $W$, which can range from $10^5$ to $10^7$.

After the training, the neurons in the hidden layer will contain the weights necessary to represent each word. Simply multiplying by the one-hot vector is enough to take as output the embedding format.

## Link prediction

Link prediction has been widely employed across various domains to estimate the likelihood of missing (or future) connections between two nodes [48]. Several techniques rely on

quantifying the similarity between nodes, which is subsequently utilized for prediction purposes [48]. In [10], weighted and unweighted local similarities are compared to predict the likelihood of future citation between two researchers. In this context, higher similarity scores indicate higher probabilities for establishing a citation link.

Another illustrative instance can be found in [49], where the Node2Vec technique is employed to encode graphs into node embeddings. Subsequently, a classification machine learning model, namely LightGBM [50], is utilized to predict edges connecting human phenotypes and genes.

In this study, we approached the link prediction task in a similar manner. We utilized the node embeddings generated in the previous step to assess the similarity between nodes. After obtaining the embeddings, we computed the cosine similarity for the edges present in the labeled dataset. For simplicity, we will henceforth use the term *walk similarity* to refer to the cosine similarity between embedding vectors produced by a specific random walk. This dataset comprises two main components: the positive edges sample, which accounts for 25% of the removed edges, and an equal number of non-edges in the form of our negative edges sample. Cosine similarity is a commonly used metric that ranges from -1 to +1, with higher values indicating a higher likelihood of connectivity between the two nodes, while lower values indicate a lack of connection. In this study, we applied a normalization procedure to scale the walk similarities between 0 and 1. The normalization was necessary since different walks produced different node similarity ranges.

## Results

### Quality comparison

Regarding the link prediction task, the primary objective of our study was to investigate whether various random walks could lead to different outcomes. To accomplish this, for each graph we used the labeled dataset containing the positive and negative edges instances and the nine similarities, one for each random walk type. Comparing the similarity with its label made it possible to determine the area under the ROC curve (AUC) and the area under the Precision-Recall Curve (AUC PR).

Both quality metrics do not require fixing a similarity threshold to establish whether a given similarity indicates an edge or non-edge prediction. Consequently, each graph yielded nine distinct performances, one for each walk.

The results of our study were presented in two ways. First, in Table 2, we arranged the median quality metrics for each walk in order of AUC. The median quality is computed across all 37 networks of the dataset.

A preliminary analysis of the results presented in Table 2 reveals proximity in the performance scores of the tested random walk strategies. Specifically, the TSAW obtained the highest AUC-ROC score at 0.864, while the Node2Vec algorithm with parameters $p = 0.5$ and $q = 1.5$ exhibited the lowest score at 0.84, with a difference of 0.024 (less than 3%). Furthermore, the DG walk achieved the AUC-PR value of 0.860, while Node2Vec(0.5, 1.5) scored 0.825, indicating a tolerable difference of 0.035. Notably, the percentage differences between the highest and lowest scores were minimal at 3% and 4%, respectively.

This is surprising, given that different random walk strategies lead to a very close performance score suggesting that changing the walk bias does not significantly affect the link prediction outcomes. A further examination of the top four performers revealed that TSAW and N2V(1.5, 0.5), occupying the first and second positions, exhibit predominantly exploratory behavior. N2V(2.0, 1.5) was ranked third, a hybrid walk, while the degree-based walk (DG), which is biased towards hubs and well-connected nodes was ranked fourth.

**Table 2. Median quality metrics for the link prediction task within 37 graphs, ordered in descending order by AUC.** The metrics utilized were the area under the ROC curve (AUC) and the area under the Precision-Recall Curve (AUC PR).

| Walk Type | AUC | AUC PR |
|---|---|---|
| TSAW | 0.864 | 0.845 |
| N2V(1.5, 0.5) | 0.864 | 0.859 |
| N2V(2.0, 1.5) | 0.861 | 0.842 |
| DG | 0.861 | 0.860 |
| N2V(0.25, 0.5) | 0.857 | 0.840 |
| N2V(1, 1) | 0.856 | 0.839 |
| RW | 0.855 | 0.841 |
| ID | 0.850 | 0.833 |
| N2V(0.5, 1.5) | 0.840 | 0.825 |

When examining the two worst-performing strategies, we can observe N2V(0.5, 1.5) with a purely local behavior and the inverse-degree walk (ID) that prioritizes isolated nodes over hubs. We can observe a trend as the best-performing strategies were those capable of exploring the network in depth, avoiding isolated nodes, and aiming to visit new and well-connected nodes. This approach had a higher probability of mapping relationships between nodes and achieving better link prediction performances. Conversely, strategies with local behavior that avoided exploring new nodes or hubs and prioritized isolated nodes yielded poorer results when compared to other strategies. We also observe that N2V(1, 1) and RW ranked sixth and seventh, respectively, with very similar results and a difference of only 0.001 and 0.002 in AUC and AUC-PR, respectively. This finding is expected, as both strategies have the same underlying setup. In Fig 4 we illustrate the performance metrics in box plots for each type of random walk ordered by the best median AUC ROC. The background lines represent each graph, and to distinguish them, we used colors ranging from purple to light orange based on their respective median AUC, from best to worst. The lines connecting the dots provide a view of how much performance changes when transitioning from one type of walking to another.

The results show that the performance outcomes are predominantly influenced by the underlying graph structure rather than the specific random walk employed. We observe that the lines exhibit a consistent lack of significant deviations across different random walks, leading to the preservation of the color spectrum throughout the walks. However, variations happened under certain conditions, specifically with the DG walk, where a visible increase in the slope is observed, and with the ID with a decrease. These findings align with the unexpected patterns discovered in Table 2, where the outcomes consistently exhibit similarities across various random walks.

When examining the box plots, it becomes evident that not only the medians are close one to another, but also the upper quartiles. The primary discernible disparity lies in the interquartile ranges, which vary across the different walks. The ID walk demonstrates the widest interquartile range for both AUC and AUC-PR. This indicates that while the ID walk may yield satisfactory performance outcomes for certain graphs, it can yield suboptimal results for others, thereby diminishing its overall reliability. Apart from the ID walk with its broader range, no discernible distinguishing features can be identified to differentiate between the various walks, as their distributions exhibit high similarity.

In conclusion, we found that different random walks yield varying performance outcomes. However, the disparities in terms of both AUC and AUC-PR medians are minimal, with a mere 3% and 4% difference respectively. Furthermore, upon examining the distributions of

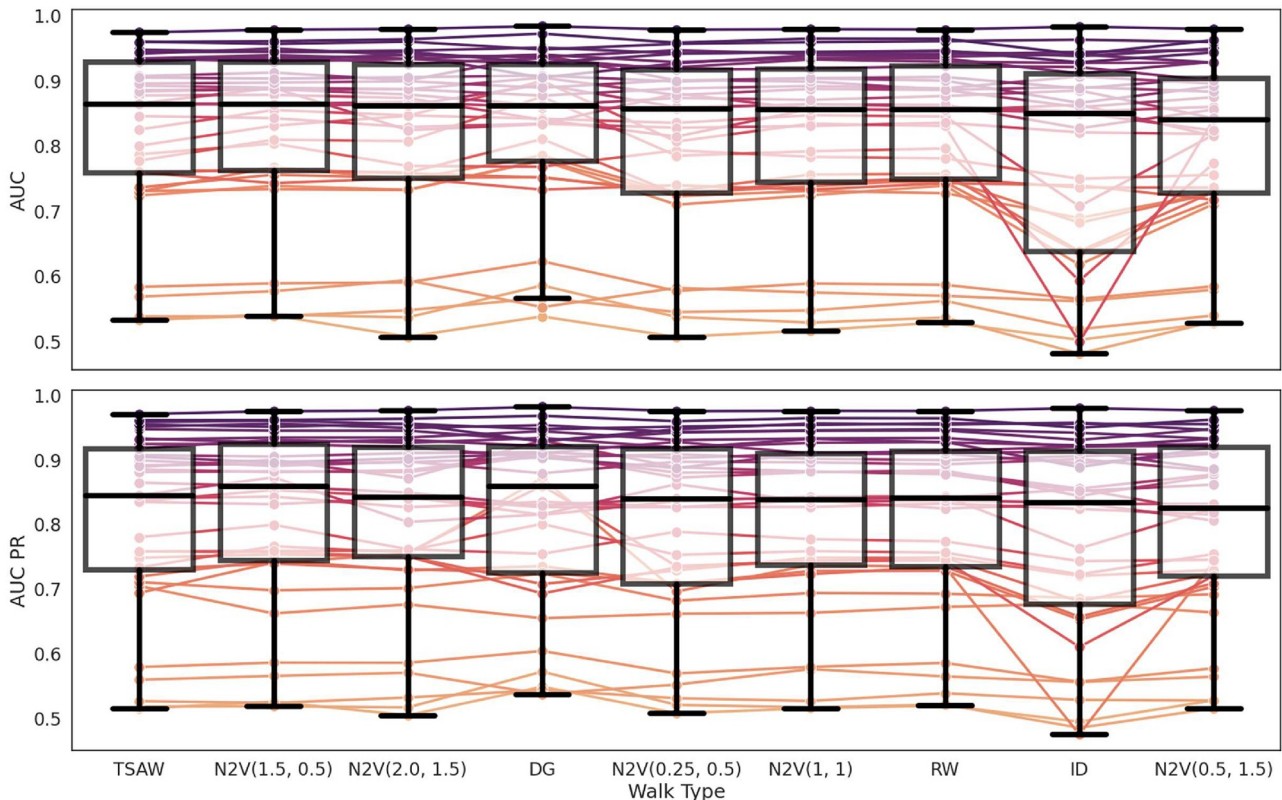

**Fig 4. Box plot of the performance obtained for the 37 networks and considering different random walks.** Each background line represents the performance of the graph itself. The lines are colorfully displayed from purple to light orange based on their respective AUC median, from the best to the worst.

the performance metrics, it becomes apparent that the medians and upper quartiles exhibit significant proximity, making it challenging to distinguish between the walks solely based on the box plot visualization. Our finding suggests that the outcomes of link prediction are more dependent on the intrinsic characteristics and properties of the network itself, rather than the specific choice of the random walk applied.

## Walk similarities correlation

In our previous analysis, we observed that distinct random walks demonstrated comparable link prediction performances. Here, we aim to probe whether diverse random walks extract different information from the underlying graph.

In the labeled dataset used in the link prediction we have nine cosine similarities for each edge (see Fig 2(d)). Based on that, for each graph was possible to calculate the Pearson correlation between the cosine similarity distributions for all combinations of walk pairs (e. g. DG vs ID, or TSAW vs N2V(1, 1)).

In Fig 5, we show two examples of correlation plots. The edges in the graphs are categorized into two groups, based on the edge label. We first show the results for the Celegans (C. elegans) Metabolic graph, where the correlation between the walks ID and DG is the lowest (0.2). In contrast, the second example in the figure shows the results for the Wiki Science graph, with the correlation between N2V(2.0, 1.5) and TSAW being be the highest (1.0).

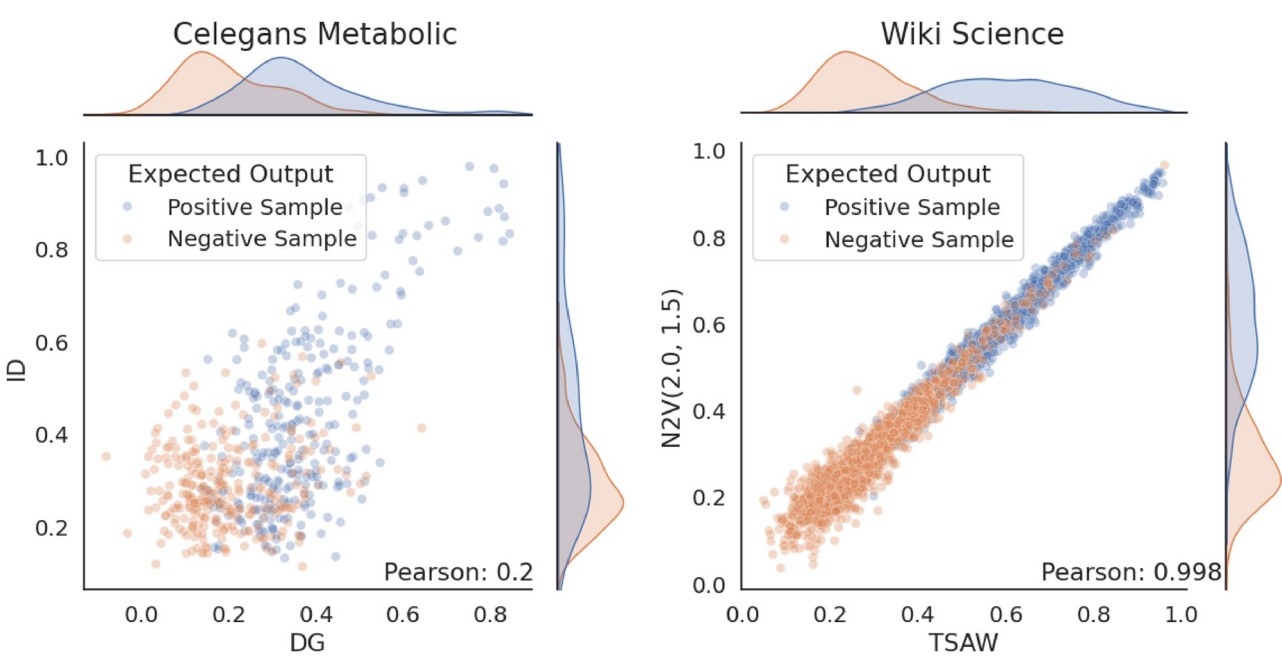

**Fig 5. Two scatter plots representing the correlations between pairs of walks.** Each figure is divided by the edges label, with the positive edges sample in blue and orange for the negative edges sample. The distribution of labels is shown on the top and right sides of each figure. In (a), we can observe the lowest correlation value of 0.2, which was observed in the Celegans (C. elegans) metabolic network between the inverse degree (ID) and the degree-based random walk (DG). Contrarily, in (b), we observe the highest correlation value of 1.0, which was observed from the Wiki Science network between Node2Vec(2.0, 1.5) and TSAW.

For the correlation, when random walks extract similar information from a network, the similarities should exhibit a proportional relationship, resulting in a distribution plot that closely resembles an identity function with a correlation value close to 1. Contrariwise, when the plot does not exhibit a clear upward or downward trend, the two random walks have low or no correlation. It is also possible to observe a negative correlation when the similarities decrease in one random walk while simultaneously increasing in the other. Consequently, we can infer that the random walks extract opposite or contrasting information from the network. The magnitude of the negative correlation, when closer to -1, indicates a stronger opposition between the extracted information.

During the conducted experiment, no pairs of random walks demonstrated this opposing behavior. This outcome aligns with our expectations, as all random walks yielded relatively good link prediction results.

When observing Fig 5, it becomes apparent that the walks exhibit distinct expected output histograms in the Celegans network. For the DG walk, two separate peaks can still be identified. However, in the ID walk, the distributions overlap, resulting in a negative impact on the discrimination power.

In contrast, in the Wiki Science network, histograms display comparable shapes for both walks, while their distributions exhibit a substantial degree of separation, resulting in a limited intersection area. This divergence in distributions contributes to enhancing the discrimination and thereby improves the accuracy in discriminating both classes.

In order to summarize the results, we generated a heatmap (see Fig 6) that depicts the median correlations between pairs of random walks. The figure reveals that the random walks exhibit strong values of correlation, with a minimum correlation coefficient of 0.85, which is

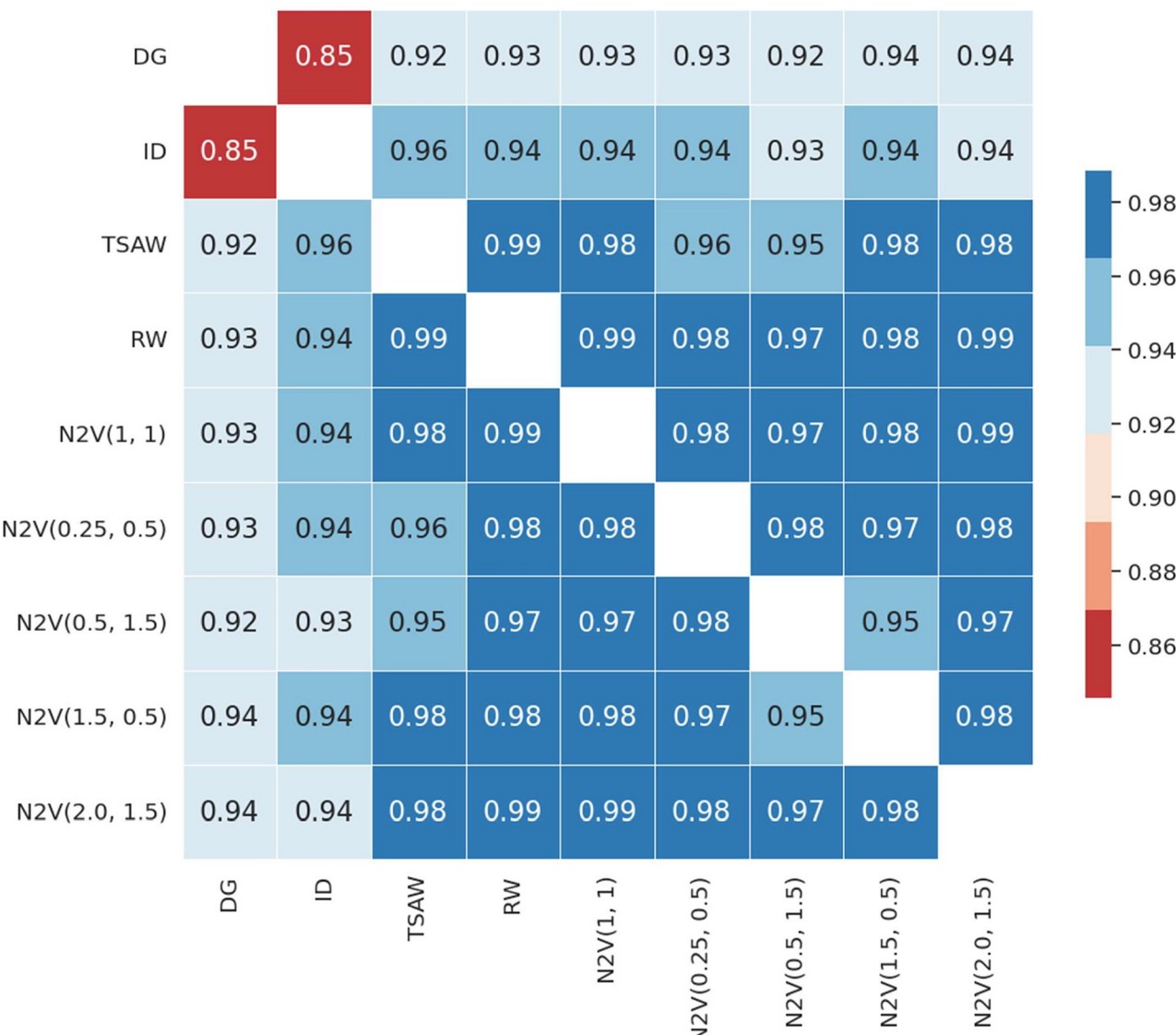

**Fig 6. Median of the Pearson correlation distribution between the random walk pairs from the 37 graphs.** This figure summarizes the distribution of the correlations as presented for the minimum and maximum correlation on Fig 5.

considered to be a high value and a maximum correlation coefficient of 0.99. The lowest correlation is observed when pairing the DG and ID walks, which is expected since these walks exhibit opposite behaviors. Moreover, DG and ID walks consistently exhibit lower correlations when paired with other walks, which is also expected since they are the only degree-based walks, contrasting with the other walks that incorporate other types of information. The DG walk achieves a maximum correlation of 0.94 when paired with the N2V(1.5, 0.5) and N2V(2.0, 1.5) walks, representing an explorer and a hybrid behavior walk, respectively. This result indicates that the DG walk displays a walk style more akin to the explorer when seeking hubs.

We achieved the maximum correlation of 0.96 when pairing TSAW and ID. This finding suggests that despite employing diverse walk styles, the retrieved information may exhibit a high degree of correlation. Table 2 illustrates that ID achieved the second lowest AUC, with a

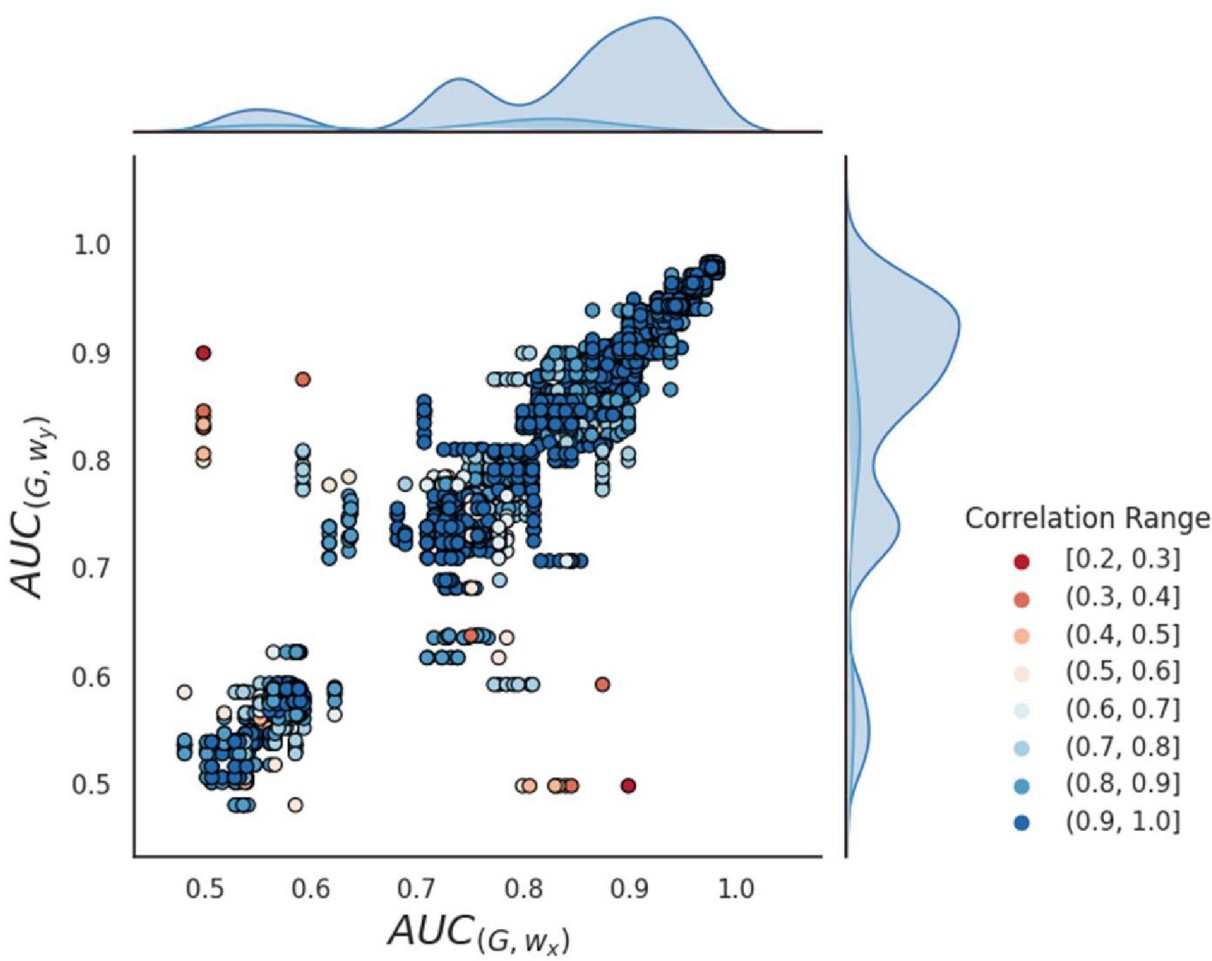

**Fig 7. AUC values for each graph $G$ and pair of random walks $w_x$ and $w_y$.** The dots on the plot are color-coded according to the Pearson correlation between the walks.

value of 0.850, whereas TSAW attained the highest AUC of 0.864. This observation is particularly interesting as it accentuates the notion that even highly correlated walks can yield disparate link prediction performances.

Excluding DG and ID from the analysis, we observe that the TSAW achieved its lowest performance when paired with N2V(0.5, 1.5) and N2V(0.25, 0.5), corresponding to a hybrid and a local walk, respectively, resulting in correlation coefficients of 0.95 and 0.96. Intriguingly, TSAW, being a memory-wise variant of RW, exhibited the highest correlation of 0.99. Hence, even though RW is part of the group with the lowest prediction performances and TSAW the best, they still have similar similarity distributions.

All Node2Vec walks exhibited very correlated distributions among themselves. With the lowest correlation of 0.95 between the two contrasting walks, N2V(1.5, 0.5) and N2V(0.5, 1.5), and the highest of 0.99 between N2V(1, 1) and RW.

Fig 7 shows the scatter plot of the AUC performance for each pair of distinct walks denoted as $w_x$ and $w_y$. The color scheme in the plot reflects the range of correlations observed between the two walks. The results validate the observations depicted in Fig 5, where walks exhibiting higher correlations tend to demonstrate similar performance, while those with lower

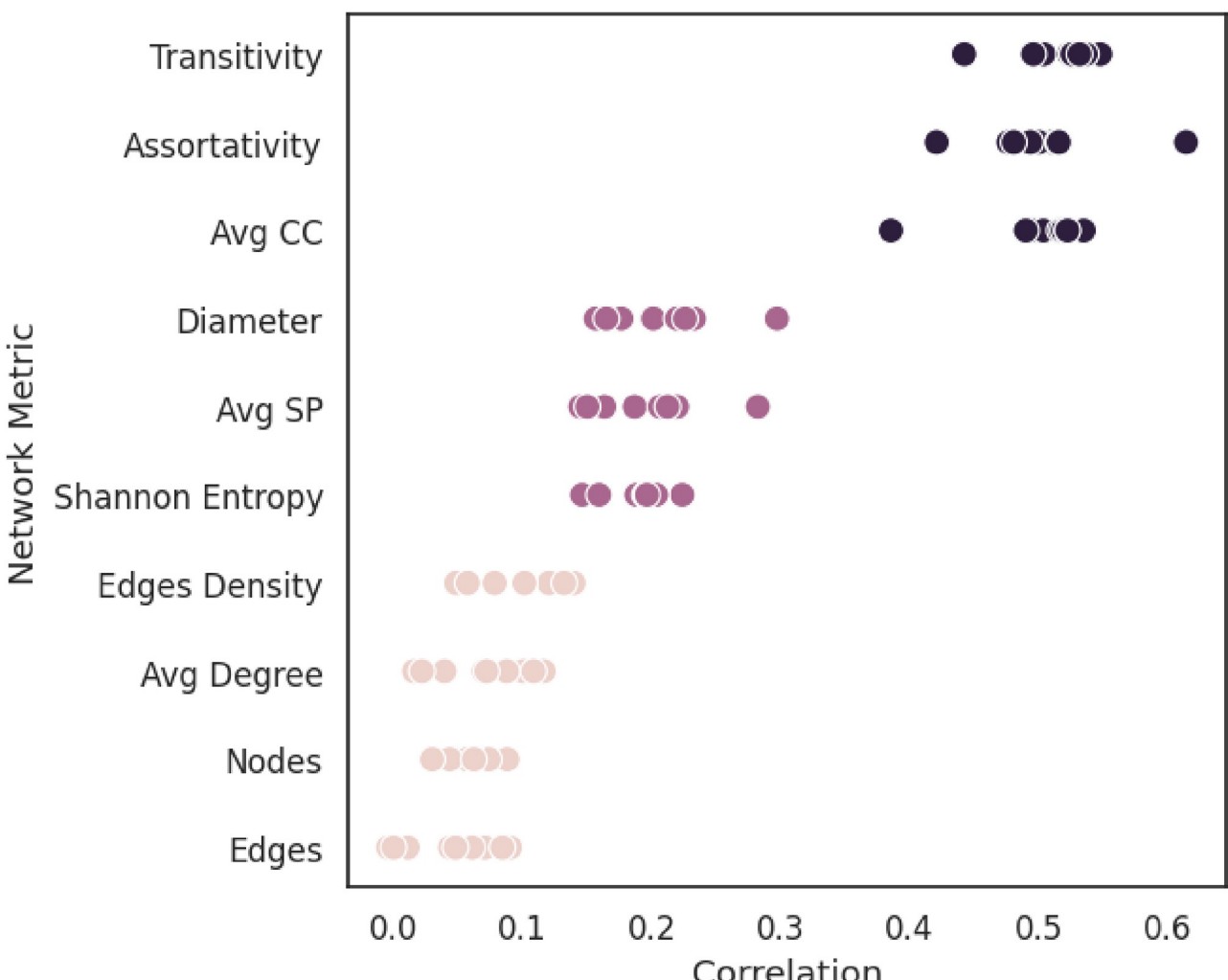

**Fig 8. Correlation between prediction performance (in terms of AUC) and network properties.**

correlations do not. This pattern is visually evident in the plot, wherein highly correlated walks display a linear relationship resembling an identity function, while walks with lower correlations appear scattered. Furthermore, the histograms provide additional insights by illustrating that the occurrences of high correlations are predominantly concentrated within the third peak, corresponding to the best AUC values ranging from 0.85 to 1. Consequently, in graphs exhibiting outstanding performance, random walks demonstrate correlation and converge toward comparable outcomes. Conversely, for graphs with low correlated walks, results vary among the walks.

To check if there are cases in which network properties seem to impact the performance of link prediction, we conducted a detailed analysis of the correlation between link prediction performance and various network topology metrics across the considered networks. Link prediction performance was measured using the area under the ROC curve (AUC ROC), and network topology was assessed using ten different metrics. As shown in the Fig 8, we identified three main groups of metrics with varying levels of correlation to prediction performance. Metrics related to basic network structures, such as node and edge counts, showed low or no

correlation (-0.004 to 0.14). Metrics reflecting network randomness, like Shannon entropy and diameter, exhibited moderate correlation (0.14 to 0.3). The highest correlation (0.38 to 0.61) was observed with metrics related to local network structures, such as assortativity and clustering coefficient. These findings indicate that certain network properties can make the problem of link prediction easier or harder, with local network characteristics playing a significant role in the prediction accuracy.

Overall, we can conclude our second research question knowing that different walks do extract very similar information, even the lowest median similarity was 0.85 and the maximum 0.99. We also observed that can see that walks with similar styles, explorer or local, tend to be more correlated among themselves rather than the others. Also, walks based on node degree, like DG and ID, still very correlated with the other walks and extracted similar information, however, had the lowest correlations. And finalizing, we saw that for graphs where the walks are very correlated, results tend to be similar, and the higher the correlation higher the changes to have a good performance.

## Conclusion

This research study aimed to analyze the behavior of various embedding information with respect to different node sequence generations in the context of link prediction. The investigation involved the examination of four conventional random walks and node2vec with different parameter settings on 37 networks. The study sought to answer two primary research questions: (1) the extent to which the performance of different random walks varies in embedding models and downstream tasks, (2) the nature of information captured by different walks, specifically concerning node similarities in link prediction.

The results revealed that there was minimal variation in the performances of different random walks for the link prediction. The median link prediction performance exhibited a difference of merely 3% to 4% in terms of AUC and AUC PR. Moreover, we also observed that random walks characterized by exploratory behavior tended to yield better results than local walks. Two explorer walks, TSAW and N2V(1.5, 0.5), achieved the highest AUC performance of 0.864, while the lowest performance was obtained by ID and N2V(0.5, 1.5), two local walks, with AUC values of 0.85 and 0.85, respectively.

Pertaining to the nature of the extracted information, our analysis of link prediction revealed a robust positive Pearson correlation between node similarities derived from different walks. This correlation held strong even when comparing node similarities obtained from fundamentally opposed walks, such as DG and ID, where the lowest median correlation across networks was as high as 0.85.

In essence, our study underscores that distinct random walks exhibit remarkably similar performance in embedding models for link prediction tasks and capture largely consistent information about node similarities. These insights hold significant implications for practical applications, particularly in scenarios where only the sequences (random walks) from a dataset are known. Based on the dataset and random walks employed, our results suggest it is possible to reconstruct network links irrespective of the specific walking type used. This opens the door for future research to test the hypothesis in a more systematic way that different random walk approaches may yield similar results in predicting network structure.

Our research has unveiled significant findings concerning the performance of various random walks and their ability to extract node similarity information from graph embeddings. However, there remains ample room for exploration. Future studies could examine the behavior and performance of random walks and embeddings in a wider array of tasks beyond link

prediction. This could include tasks such as community detection, anomaly detection, or network evolution modeling.

Lastly, it would be intriguing to probe deeper into the nature of information captured by different walks. For instance, future work could explore the kinds of structural or topological properties that different walks are more sensitive to and how these properties impact downstream tasks. This could offer insights into how to choose or design the most suitable walk or embedding technique for specific network types or tasks.

## Author Contributions

**Conceptualization:** Adilson Vital, Jr., Filipi Nascimento Silva, Diego Raphael Amancio.

**Data curation:** Adilson Vital, Jr.

**Investigation:** Adilson Vital, Jr., Filipi Nascimento Silva.

**Methodology:** Adilson Vital, Jr., Filipi Nascimento Silva, Diego Raphael Amancio.

**Project administration:** Diego Raphael Amancio.

**Software:** Adilson Vital, Jr.

**Supervision:** Filipi Nascimento Silva, Diego Raphael Amancio.

**Validation:** Adilson Vital, Jr.

**Visualization:** Adilson Vital, Jr.

**Writing – original draft:** Adilson Vital, Jr., Filipi Nascimento Silva, Diego Raphael Amancio.

**Writing – review & editing:** Adilson Vital, Jr., Filipi Nascimento Silva, Diego Raphael Amancio.

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
