## [Decision Letter · Decision Letter 0]

24 Apr 2024

PONE-D-23-36575Comparing biased random walks in graph embedding and link predictionPLOS ONE

Dear Dr. Silva,

Thank you for submitting your manuscript to PLOS ONE. After careful consideration, we feel that it has merit but does not fully meet PLOS ONE’s publication criteria as it currently stands. Therefore, we invite you to submit a revised version of the manuscript that addresses the points raised during the review process.

You will see that both reviews recommend that you revise your manuscript, so please consider making the suggested changes. After my own reading of the manuscript, I agree with them in that the paper is interesting, well-written in general and has potential for publication after a major revision. Please consider making the suggested changes to better highlight the contribution of your work. If you feel you can comprehensively address the reviewer's concerns, please provide a point-by-point response to these comments along with your revision. Please show all changes in the manuscript text file with track changes or color highlighting. If you are unable to address specific reviewer requests or find any points invalid, please explain why in the point-by-point response.

We look forward to receiving your revised manuscript.

Kind regards,

Pablo Martin Rodriguez

Academic Editor

PLOS ONE

Journal Requirements:

Additional Editor Comments (if provided):

Reviewers' comments:

Reviewer's Responses to Questions

**Comments to the Author**

1. Is the manuscript technically sound, and do the data support the conclusions?

Reviewer #1: Yes

Reviewer #2: Yes

2. Has the statistical analysis been performed appropriately and rigorously? 

Reviewer #1: Yes

Reviewer #2: Yes

3. Have the authors made all data underlying the findings in their manuscript fully available?

Reviewer #1: Yes

Reviewer #2: No

4. Is the manuscript presented in an intelligible fashion and written in standard English?

Reviewer #1: Yes

Reviewer #2: Yes

5. Review Comments to the Author

Reviewer #1: The authors compared how different types of random walks impact link prediction tasks. The paper is well-organized and well-written. However, I still have a few major concerns that I listed below:

1) Throughout the paper, the authors mention that they are comparing biased random walks. However, the traditional random walk is not biased; in some parts of the text, it is referred to as a biased random walk. Note also that P(v,u) does not depend on v, but only on u. For instance, see the title and the first paragraph of the Methodology section, "We leveraged nine biased random walk algorithms ..."

2) Although enough for the paper's purposes, The definition in the Section "Random walk" is not precise. For instance, the random walk is defined as a "sequence of nodes traversed during a walk through a network." However, this is the definition of a finite walk. To be a random walk, you have to ensure that the sequence is random. Also, the random walk can be infinite. Being finite is not a necessary condition.

3) In the true self-avoiding random walk on page 8, the vector f_x is defined as the "frequency of visits to x." How do you define the frequency? Would it be the number of times the random walker passes through node x divided by the total number of steps? After equation (5), it seems to be the number of visits. If I understood correctly, both definitions should lead to the same transition probability, but it would be better to clarify this point to make the paper more consistent and easier to read.

4) The Node2Vec description is not clear. Perhaps my questions are due to the fact that I am not so familiar with this random walk, but it would be better to clarify this to make the paper more accessible to a wider audience. My main question here is how to write this random walk transition probability in the form of Eq. (1). Since you have two parameters (p and q), it is not clear how the transition probabilities sum up to one, i.e., \\sum_v P(u,v) = 1 for a non-lazy random walk.

5) My main concern about the paper is the importance and impact of the paper's results. In the last paragraph of the "Quality Comparison" section, the authors mention: "Our finding suggests that the outcomes of link prediction are more dependent on the intrinsic characteristics and

properties of the network itself, rather than the specific choice of the random walk applied." However, this seems to be a trivial result. Moreover, this makes us ask more questions, for example, what are the features that make it easier/harder to improve the results? One possibility would be to test for the impact of correlations, which could be tested by doing a similar experiment using the configuration model as a null model, generating an uncorrelated network, and comparing the quality of the original and the uncorrelated version (maybe for a few networks).

6) The database. Most of the analyzed networks are relatively small. So, what is the impact of the network sizes on the quality of the predictions? Could it be that the results change for larger networks? The authors should clarify that or provide evidence that their results should hold for larger networks.

7) The walk similarities correlations should be better described. I could not understand the analysis very well. For example, in Fig. 5b, at the bottom, there is "Pearson 1.0," however, this coefficient does not refer to how the points in the plot are correlated. Note that a Pearson = 1.0 should be a perfect straight line.

Minor details:

a) On page 7, when defining the maximum walk length, the authors mention, "in this project." Perhaps substituting the word project for paper would be slightly better.

b) There is a typo in Eq. (6), which is defined using the Greek letter rho, while the letter p is used in the following paragraphs.

c) After Eq. (7), there is a paragraph space that should be removed.

d) Fig. 4 is a bit confusing. The plot is continuous, but the x-axis is discrete. I understand that this is done for visualization purposes, but we do not know which network is represented by each color, and it is still difficult to follow the lines. Perhaps just the points would be enough.

In general, the authors should strengthen their results by emphasizing the importance of their findings and how they can be helpful for future research. Points 5 - 7 are my major concerns, while points 1-4 are important definitions that can be easily fixed. Thus, I would suggest a major revision.

Reviewer #2: The paper addresses the effect of the choice of the random walk for graph embedding. Two

main experiments are performed: 1) the ability of the embedding to predict the

missing edges in the graph; and 2) the similarity of the embeddings (in terms of the

correlation of cosine similarities of the endpoints of the edges).

The subject is interesting and the paper is well written. The experiments are

very objective and the results are clear. Paper organization is appropriate.

I agree with most of the conclusions of the paper. However, authors state that their

results "suggest that we can potentially recover the underlying network structures from

such data, regardless of the nature of the walks performed, adding to the versatility of

network science in analyzing complex systems." I especially disagree with this statement.

Although different random walks lead to similar performance in link prediction and the

embeddings share some similarities, the leap to the conclusion that the underlying network

structure can be recovered from the embeddings independently of the random walk seems too

strong. I suggest to tone down this statement; maybe stating as a hypothesis for future

works.

For me a major issue is the lack of information about model training setup. Authors state

"hyperparameters of the embedding model were optimized to ensure optimal and consistent

performance." What does that even mean? How can you guarantee optimality? What is

consistency here? Please provide more details about the training setup.

There are also minor issues that should be addressed:

- The way authors talk about the random walk mechanism in the node2vec algorithm is

sometimes confusing: "different configurations of node2vec." Actually, they parametrize

the underlying random walk in node2vec with five different settings. I suggest to

clarify this.

- node2vec is sometimes written as "Node2Vec" and sometimes as "node2vec". Please

standardize the notation.

- Authors use notation Log, log, and ln. Please standardize and clarify the notation.

- p. 2, "Our dataset revealed" -> "Our experiments revealed"

- Sentence "This becomes particularly insightful for datasets such as textual..." in the

Introduction is not clear.

- When explaining second-order proximity (about LINE), what does "similarity between p_u

and p_v" mean mathematically?

- Authors state that ID "outperforms other degree-biased walks in terms of time

efficiency." What does this mean? In a similar vein, what does

"TSAW achiving the best learning curve" mean?

- For me N2V(1, 1) *is* RW. Why test both? Is it because the different embedding

strategies?

- What is the "algorithm equipped with a built-in function"?

- What is \\rho in eq. (6)?

- In the sentence "we applied normalization procedure to scale the walk similarities

between 0 and 1", what is walk similarity?

Some minor comments:

- For me, 25% of missing edges is a lot. Is there any reason for this choice?

- Just a matter of taste, but I think eq. (2), (3), (4) and (5) could be written for

\\tau_uv instead of P(v|u). This way you make use of eq. (1) and the verbose

denominators are avoided.

- Is the choice of \\alpha = 200 appropriate for small networks (n < 200)? I am not sure

about this.

6. PLOS authors have the option to publish the peer review history of their article (what does this mean?). If published, this will include your full peer review and any attached files.

Reviewer #1: No

Reviewer #2: No

---

## [Author Response · Author response to Decision Letter 0]

26 Jul 2024

See response attached with figures.

Thank you for your insightful comments on our manuscript. We appreciate the thorough reviews and the opportunity to improve our work based on your feedback. Below, we address each of the major and minor concerns raised by the reviewers.

Reviewer #1: The authors compared how different types of random walks impact link prediction tasks. The paper is well-organized and well-written. However, I still have a few major concerns that I listed below:

1) Throughout the paper, the authors mention that they are comparing biased random walks. However, the traditional random walk is not biased; in some parts of the text, it is referred to as a biased random walk. Note also that P(v,u) does not depend on v, but only on u. For instance, see the title and the first paragraph of the Methodology section, "We leveraged nine biased random walk algorithms ..."

Response: We agree that the traditional random walk and the Node2Vec with parameters p and q set to 1, which rely solely on the information from the origin, can not be considered biased. As suggested, we revised the manuscript to clearly distinguish between these two cases and adjust the equation accordingly.

2) Although enough for the paper's purposes, The definition in the Section "Random walk" is not precise. For instance, the random walk is defined as a "sequence of nodes traversed during a walk through a network." However, this is the definition of a finite walk. To be a random walk, you have to ensure that the sequence is random. Also, the random walk can be infinite. Being finite is not a necessary condition.

Response: We have corrected the definition of random walk using the following sentences: "A random walk is a sequence of nodes visited during a random traversal of a network. While this paper focuses on random walks with a finite number of steps, in general, the sequence of nodes can be infinite."

3) In the true self-avoiding random walk on page 8, the vector f_x is defined as the "frequency of visits to x." How do you define the frequency? Would it be the number of times the random walker passes through node x divided by the total number of steps? After equation (5), it seems to be the number of visits. If I understood correctly, both definitions should lead to the same transition probability, but it would be better to clarify this point to make the paper more consistent and easier to read.

Response: Frequency refers to the absolute number of times a specific node was visited. Both definitions, in fact, result in the same transition probability. This has been clarified in the revised version of the manuscript:

"where $f_{x}$ refers to the absolute number of times node $x$ has been visited."

4) The Node2Vec description is not clear. Perhaps my questions are due to the fact that I am not so familiar with this random walk, but it would be better to clarify this to make the paper more accessible to a wider audience. My main question here is how to write this random walk transition probability in the form of Eq. (1). Since you have two parameters (p and q), it is not clear how the transition probabilities sum up to one, i.e., \\sum_v P(u,v) = 1 for a non-lazy random walk.

Response: Node2Vec's transition mechanism is based on weighted edges, where the weights influence the likelihood of moving from one node to another. The sum of these weights leaving a node is not necessarily 1, but they can be normalized to represent probabilities. While it is possible to convert these weights into a probability function similar to Equation (1), doing so would add complexity for readers. This is because Node2Vec's walk behavior fall into three distinct categories: (1) returning to the immediately previous node, (2) moving to a node that is a neighbor of both the current and previous nodes, and (3) moving to any other node connected to the current one. This weighting scheme allows Node2Vec to effectively explore both local and global network structures. For clarity and ease of understanding, we chose to use closed formulas like Equation (1) whenever possible. However, for Node2Vec, we decided to retain the original weighted representation.

5) My main concern about the paper is the importance and impact of the paper's results. In the last paragraph of the "Quality Comparison" section, the authors mention: "Our finding suggests that the outcomes of link prediction are more dependent on the intrinsic characteristics and

properties of the network itself, rather than the specific choice of the random walk applied." However, this seems to be a trivial result. Moreover, this makes us ask more questions, for example, what are the features that make it easier/harder to improve the results? One possibility would be to test for the impact of correlations, which could be tested by doing a similar experiment using the configuration model as a null model, generating an uncorrelated network, and comparing the quality of the original and the uncorrelated version (maybe for a few networks).

Response:

We respectfully disagree with the assertion that our result is trivial. The sequences generated by different types of random walks vary significantly, displaying biases such as visiting hubs more frequently, skipping certain edges, or avoiding repeating the same path, depending on the walk dynamics. These variations imply that the sequences perceive the network in distinct ways. This distinction is crucial in the node2vec approach, which relies on the sequences from the walks to capture different structural and functional aspects of the network, in particular the original proposal of the algorithm suggests the use of two different random walk dynamics to capture different information from the network (e.g., global vs local). For that reason, the impact of network topology versus the chosen dynamics cannot be considered trivial without empirical validation. Unless there is existing literature that establishes this as a triviality, our findings provide valuable insights.

We agree with the reviewer on the importance of investigating the impact of different network characteristics. Instead of using potentially unrealistic random models to control network features, we opted to analyze our existing diverse set of networks. This approach allows us to evaluate the effects of intrinsic network properties in a more realistic setting.

For each of the nine random walks, we calculated the correlation between link prediction performance and various network topology metrics across 37 networks. Link prediction performance was measured using the area under the ROC curve (AUC ROC), and network topology was assessed using ten different metrics, ranging from basic metrics like the number of nodes and edges to more sophisticated ones such as entropy, transitivity, and assortativity.

As depicted in the figure below, our results suggest three main groups of metrics with varying levels of correlation:

 1. Low or No Correlation (-0.004 to 0.14): This group includes metrics that are weakly correlated with prediction performance. This group comprises metrics that define simple network structures, such as node and edge counts, average node degree, and edge density (the current number of edges divided by the total possible). This means that besides they can be important to describe the network’s dimensionalities they have a low correlation with predictions’ performances.

 2. Moderate Correlation (0.14 to 0.3): Consists of three metrics: Shannon entropy, diameter, and average shortest path. Shannon entropy measures how disordered and random a network’s connections are, diameter represents the greatest distance between any two nodes in a network, and the average shortest path indicates the mean of the minimum distances between all nodes. In real networks, due to the Power Law effect, well-connected nodes tend to receive more connections as the network grows, creating clusters and reducing the minimum distances among nodes. However, in random networks, as the network grows, distances also grow, leading to larger shortest paths and diameters. The moderate correlation suggests that while randomness characteristics play a role in the predictions, they are not sufficiently influential for a high correlation. This makes sense, as more complex or random network patterns are harder to understand and predict missing connections.

 3. Relatively High Correlation (0.38 to 0.61): Metrics in this group transitivity, assortativity, and the average clustering coefficient. Transitivity measures the probability that the neighbors of a node are connected. Assortativity, a local metric, indicates how nodes with similar degrees are connected, and the average clustering coefficient reflects the tendency of nodes to form clusters. High transitivity and clustering coefficient values suggest that networks with these characteristics are easier for a classification algorithm to predict missing edges within clusters. Similarly, highly assortative networks have a regular and predictable pattern, as nodes with similar degrees tend to be connected.

Overall, metrics related to local network structures, such as assortativity, transitivity, and clustering coefficient, correlate highly with better link prediction performance. These metrics indicate that networks with regular and predictable structures are more conducive to accurate link predictions. On the other hand, metrics associated with randomness characteristics showed moderate correlations, suggesting that while they increase forecasting complexity, they still play a role in link prediction performance. Basic network metrics like the number of nodes and edges have almost zero correlation with performance.

We have included the following text to the manuscript:

"To check if there are cases in which network properties seem to impact the performance of link prediction, we conducted a detailed analysis of the correlation between link prediction performance and various network topology metrics across the considered networks. Link prediction performance was measured using the area under the ROC curve (AUC ROC), and network topology was assessed using ten different metrics. As shown in the figure~\\ref{figure:correlation_properties}, we identified three main groups of metrics with varying levels of correlation to prediction performance. Metrics related to basic network structures, such as node and edge counts, showed low or no correlation (-0.004 to 0.14). Metrics reflecting network randomness, like Shannon entropy and diameter, exhibited moderate correlation (0.14 to 0.3). The highest correlation (0.38 to 0.61) was observed with metrics related to local network structures, such as assortativity and clustering coefficient. These findings indicate that certain network properties can make the problem of link prediction easier or harder, with local network characteristics playing a significant role in the prediction accuracy."

6) The database. Most of the analyzed networks are relatively small. So, what is the impact of the network sizes on the quality of the predictions? Could it be that the results change for larger networks? The authors should clarify that or provide evidence that their results should hold for larger networks. 

Response: Plotting the area under the ROC curve (AUC) versus network size, we can see that most graphs with up to 1000 nodes have an AUC ranging between 0.75 and 0.95. Besides that, small networks with less than 500 nodes underperformed, with an AUC between 0.5 and 0.6, while relatively large graphs with more than 1000 nodes performed similarly to the first group. We can infer that the network size is not a determining factor for a good prediction performance of the dataset utilized. Besides that, small networks can perform poorly which can be explained by the same random walk pass multiple times through the same node sequence, causing overfitting when generating the embeddings, and resulting in low-quality vectors.

7) The walk similarities correlations should be better described. I could not understand the analysis very well. For example, in Fig. 5b, at the bottom, there is "Pearson 1.0," however, this coefficient does not refer to how the points in the plot are correlated. Note that a Pearson = 1.0 should be a perfect straight line.

Response: We have corrected the plot by adding 3 decimals to the values, so we have 0.998 of Pearson correlation when comparing N2V(2.0, 1.5) versus TSAW for the Wiki Science network. We also adjusted the transparency of the dots to better visualize overlapping nodes between the two classes and to highlight areas with higher dot density along the identity line.

Minor details:

a) On page 7, when defining the maximum walk length, the authors mention, "in this project." Perhaps substituting the word project for paper would be slightly better.

Response: Done.

b) There is a typo in Eq. (6), which is defined using the Greek letter rho, while the letter p is used in the following paragraphs.

Response: Done.

c) After Eq. (7), there is a paragraph space that should be removed.

Response: Done.

d) Fig. 4 is a bit confusing. The plot is continuous, but the x-axis is discrete. I understand that this is done for visualization purposes, but we do not know which network is represented by each color, and it is still difficult to follow the lines. Perhaps just the points would be enough.

Response: We understand that the overlapping plots (boxplot versus line plot) might be confusing. However, the goal is to show that the performances for each graph, distinguished by colors, remain stable across different random walks, with significant changes occurring only in/out of DG and ID. While we acknowledge that dots would simplify the visualization and improve readability, it would compromise our initial objective of observing potential variations. Below, we present a comparison of the two plots, highlighting how the line plot better captures these variations. We have added the following information to the context.

“In Figure \\ref{figure:boxplot} we illustrate the performance metrics in box plots for each type of random walk ordered by the best median AUC ROC. The background lines represent each graph, and to distinguish them, we used colors ranging from purple to light orange based on their respective median AUC, from best to worst. The lines connecting the dots provide a view of how much performance changes when transitioning from one type of walking to another.”

In general, the authors should strengthen their results by emphasizing the importance of their findings and how they can be helpful for future research. Points 5 - 7 are my major concerns, while points 1-4 are important definitions that can be easily fixed. Thus, I would suggest a major revision.

Response: Thank you for your suggestions to improve the manuscript.

Reviewer #2: The paper addresses the effect of the choice of the random walk for graph embedding. Two main experiments are performed: 1) the ability of the embedding to predict the missing edges in the graph; and 2) the similarity of the embeddings (in terms of the correlation of cosine similarities of the endpoints of the edges). The subject is interesting and the paper is well written. The experiments are very objective and the results are clear. Paper organization is appropriate.

I agree with most of the conclusions of the paper. However, authors state that their results "suggest that we can potentially recover the underlying network structures from

such data, regardless of the nature of the walks performed, adding to the versatility of

network science in analyzing complex systems." I especially disagree with this statement. Although different random walks lead to similar performance in link prediction and the embeddings share some similarities, the leap to the conclusion that the underlying network structure can be recovered from the embeddings independently of the random walk seems too strong. I suggest to tone down this statement; maybe stating as a hypothesis for future works.

Response: We agree with your suggestion and have revised the sentence to soften its tone:

“Based on the dataset and random walks employed, our results suggest it is possible to reconstruct network links irrespective of the specific

---

## [Editor Report · Decision Letter 1]

15 Oct 2024

Comparing random walks in graph embedding and link prediction

PONE-D-23-36575R1

Dear Dr. Silva,

We’re pleased to inform you that your manuscript has been judged scientifically suitable for publication and will be formally accepted for publication once it meets all outstanding technical requirements.

Kind regards,

Pablo Martin Rodriguez

Academic Editor

PLOS ONE
---

## [Editor Report · Acceptance letter]

28 Oct 2024

PONE-D-23-36575R1 

PLOS ONE

Dear Dr. Silva, 

I'm pleased to inform you that your manuscript has been deemed suitable for publication in PLOS ONE. Congratulations! Your manuscript is now being handed over to our production team.

Kind regards, 

on behalf of

Professor Pablo Martin Rodriguez 

Academic Editor

PLOS ONE